# Br⁻/BrO⁻-mediated highly efficient photoelectrochemical epoxidation of alkenes on α-Fe₂O₃

Yukun Zhao [1,2], Mengyu Duan[1,2], Chaoyuan Deng[1,2], Jie Yang[1,2], Sipeng Yang[1,2], Yuchao Zhang [1,2], Hua Sheng [1], Youji Li[3], Chuncheng Chen [1,2] ✉ & Jincai Zhao [1,2]

Epoxides are significant intermediates for the manufacture of pharmaceuticals and epoxy resins. In this study, we develop a Br⁻/BrO⁻ mediated photoelectrochemical epoxidation system on α-Fe₂O₃. High selectivity (up to >99%) and faradaic efficiency (up to $82 \pm 4\%$) for the epoxidation of a wide range of alkenes are achieved, with water as oxygen source, which are far beyond the most reported electrochemical and photoelectrochemical epoxidation performances. Further, we can verify that the epoxidation reaction is mediated by Br⁻/BrO⁻ route, in which Br⁻ is oxidized non-radically to BrO⁻ by an oxygen atom transfer pathway on α-Fe₂O₃, and the formed BrO⁻ in turn transfers its oxygen atom to the alkenes. The non-radical mediated characteristic and the favorable thermodynamics of the oxygen atom transfer process make the epoxidation reactions very efficient. We believe that this photoelectrochemical Br⁻/BrO⁻-mediated epoxidation provides a promising strategy for value-added production of epoxides and hydrogen.

Epoxides, which are usually obtained by epoxidation of alkenes, are important intermediates for the production of valuable chemicals, especially for the applications of pharmaceuticals and epoxy resins[1–6]. Conventionally, alkene epoxidation usually requires stoichiometric peroxide-based oxidants (m-chloroperoxybenzoic acid, PhIO, hydrogen peroxide) with strict temperature control[3,7–10], or employs noble-metal catalysts at high temperature and pressure[11,12]. The electrochemical epoxidation is regarded as a sustainable and safe route for oxygenation of alkenes under mild conditions[3,13]. However, the direct electrochemical and photoelectrochemical epoxidation of alkenes behaved badly[3,14], which are limited by poor faradaic efficiency (FE) and narrow substrate applicability. Instead of poor direct electrochemical epoxidation behaviors, the indirect electrochemical epoxidations by using redox mediators (such as Cl⁻/Cl₂, Br⁻/Br₂) have been reported to promote the performance of the electrochemical epoxidation, by using precious metals (such as Pt) as anodes[4,15–22]. For example, by coupling the heterogeneous chlorine evolution reaction

and homogeneous alkene oxidation reaction with an interface, Leow et al., recently reported the chloride-mediated electrochemical epoxidation of ethylene and propylene can reach 97% selectivity and 71% FE[4]. However, the most indirect electrochemical epoxidation by Br-based redox mediator suffered from negative effects of single-electron radicals and complex muti-step oxidation pathways (Fig. 1a), which would lead to a poor current efficiency and low selectivity for epoxidation (the FE values of most epoxides are less than 40%, Supplementary Table 1 lists some reported examples with the best performance).

Photoelectrochemical (PEC) techniques, which can utilize photogenerated holes/electrons to achieve chemical conversion and reduce greatly the consumption of electric energy compared with the pure electrochemical methods, have been extensively investigated for solar energy storage and high-value chemical synthesis[23–28]. Hematite (α-Fe₂O₃) has been extensively studied as a promising photoanode material for water oxidation during PEC water splitting for H₂

[1]Key Laboratory of Photochemistry, CAS Research/Education Center for Excellence in Molecular Sciences, Institute of Chemistry, Chinese Academy of Sciences, Beijing, P. R. China. [2]University of the Chinese Academy of Sciences, Beijing, P. R. China. [3]College of Chemistry and Chemical Engineering, Jishou University, Hunan, P. R. China. ✉e-mail: ccchen@iccas.ac.cn

**a** Br⁻/Br₂-mediated EC epoxidation on Pt (electron transfer pathway)

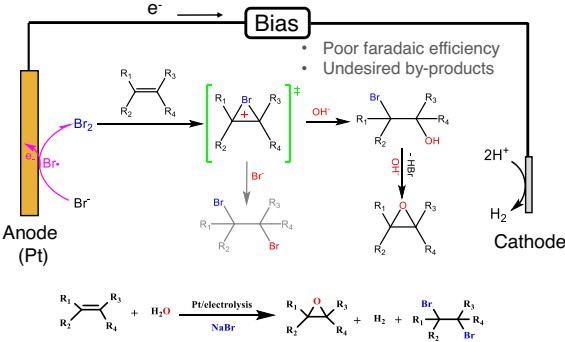

**b** Br⁻/BrO⁻-mediated PEC epoxidation on α-Fe₂O₃ (OAT pathway)

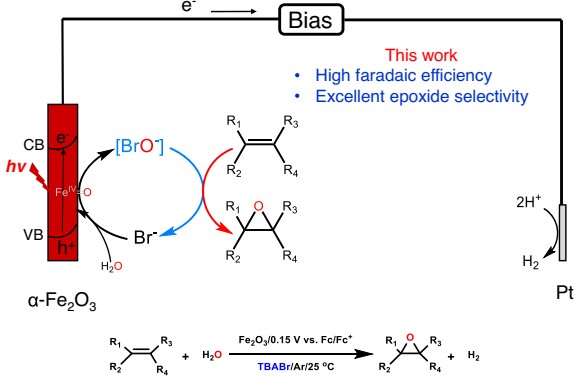

**Fig. 1 | Schematic illustrations for epoxidation of alkenes by different bromine-mediated pathways. a** Br⁻/Br₂-mediated electrochemical epoxidation on Pt anode with a pathway of single electron transfer; (**b**) Br⁻/BrO⁻-mediated photoelectrochemical epoxidation on α-Fe₂O₃ with an oxygen atom transfer pathway.

evolution. On α-Fe₂O₃ photoanodes, the surface trapped holes have been identified as high-valent iron-oxo intermediates (Fe$^{IV}$ = O). The nucleophilic attacking of water molecule on these Fe$^{IV}$ = O to form O-O bond is the key step for water oxidation[29–32]. Recently, we reported that a series of inorganic ions (such as NO₂⁻ and As$^{III}$) and organic substrates (sulfide and phosphine) with oxygen atom accepting sites can attack these Fe$^{IV}$ = O sites, and lead to oxygenation of these substrates by oxygen atom transfer (OAT) pathway[14]. However, we found that the epoxidation of alkenes, which is also a good oxygen atom acceptor, exhibits relatively poor selectivity and FE on α-Fe₂O₃. For example, the selectivity and FE for oxidation of *trans*-stilbene are only 72.1 and 11.6%, respectively[14]. On electrode with manganese oxide nanoparticles, the epoxidation of cyclooctene by oxygen transfer exhibited a FE value of 30%[3]. Very recently, it was reported that electron-rich metallic RuO₂ nanocrystals supported on nitrogen-doped carbons can boost the selective electrochemical epoxidation via direct OAT pathway. These nanocrystals exhibit excellent selectivity and FE only for the epoxidation of cyclooctene, but the epoxidation of other alkenes such as styrene is far less than satisfactory[33]. The poor performance for these direct OAT processes should be at least partially attributed to the incompatibility between the hydrophilicity of oxide surface and the low polarity of C = C bond, which is unfavorable for the interaction between the surface Fe$^{IV}$ = O sites and the alkene, and consequently hinders the direct OAT reaction.

In this work, we introduced a Br⁻/BrO⁻-mediator between α-Fe₂O₃ and alkenes to achieve highly efficient epoxidation reactions of the alkenes (Fig. 1b). A wide scope of aromatic and aliphatic alkenes was efficiently oxygenated to the corresponding epoxides, in which water is the only oxygen source and hydrogen is the product of cathode

reaction (FE > 90%). The unique characteristics of this Br⁻/BrO⁻-mediated PEC epoxidation on α-Fe₂O₃ were further verified by systematic comparison with the Br⁻/Br₂-mediated processes on TiO₂ photoanodes and Pt anodes, on which the selectivity, FE, and stability for epoxidation were rather poor. Our results provide a unique and low-energy consumption strategy for effective PEC epoxidation of different alkenes with Br⁻/BrO⁻-mediated pathway.

## Results and discussion
### Different PEC epoxidation behaviors
The alkene epoxidation reactions were conducted in a single-compartment PEC setup (Supplementary Fig. 1), with α-Fe₂O₃ as the working photoanode, Pt wire as the counter electrode and Ag/AgCl as the reference electrode. The characterizations demonstrate that α-Fe₂O₃ photoanode consists of nanorods with the diameter of 50–100 nm in the crystalline phase of hematite, which shows a good visible-light response with band gap of ~2.1 eV. The corresponding valence states of Fe and O elementals on α-Fe₂O₃ are +3 and −2, respectively (see Supplementary Figs. 2–3 for more detailed description). As depicted by linear sweep voltammetry (LSV) analysis in Fig. 2b and Supplementary Fig. 4, in acetonitrile with tetrabutylammonium tetrafluoroborate (TBABF₄) as the electrolyte and 5% H₂O (v:v) as oxygen and hydrogen sources, the onset potential was −0.02 V vs. Fc/Fc⁺, above which the water is oxidized. Interestingly, when TBABF₄ was replaced by bromide salt of TBA⁺ (tetrabutylammonium bromide, TBABr), the onset potential was lowered by about 300 mV (−0.32 V vs. Fc/Fc⁺), and the photocurrent also became much higher. It is evident that, in the TBABr system, the Br⁻ is labile and participates in the oxidization reaction on the photoanode. The lower onset potential and higher photocurrent indicate that the oxidation of bromide on α-Fe₂O₃ (Eqs. 1 or 2) is much easier than water oxidation (Eq. 3). The PEC behaviors in the presence of 4,4'-dimethyl-*trans*-stilbene (1, Fig. 2a) as the model alkene also exhibited distinct differences between TBABF₄ and TBABr systems. In TBABF₄ system, although the addition of 1 did not change the onset potential and initial photocurrent much, the photocurrent (black solid curve) was obviously enhanced at high bias (>0.15 V vs. Fc/Fc⁺) due to the oxidation of 1, which is consistent with the oxidation of methyl phenyl sulfide on α-Fe₂O₃ with TBABF₄ as electrolyte in our previous study[14]. By contrast, in TBABr system, the presence of 1 did not influence the photocurrent in the whole tested voltage range of −0.95 V to 1.05 V vs. Fc/Fc⁺ (red solid curves), which suggests that little direct oxidation of 1 occur even at high bias, since no additional current appears by adding 1. The oxidation of 1 should be mediated by species from PEC oxidation of bromide. Moreover, the difference in applied biases between the dark and light conditions is about 0.9 V to achieve the same current on α-Fe₂O₃ (Supplementary Fig. 4), indicating that the introduced photons can largely decrease the electric consumption (details in supplementary energy balance).

$$2Br^- + 2OH^- \rightarrow BrO^- + 2H_2O + 2e^- \quad E_0 = 0.76\,V \quad (1)$$

$$2Br^- \rightarrow Br_2 + 2e^- \quad E_0 = 1.07\,V \quad (2)$$

$$2H_2O \rightarrow O_2 + 4H^+ + 4e^- \quad E_0 = 1.23\,V \quad (3)$$

The Br-mediated oxidation of the alkene **1** on α-Fe₂O₃ was further carried out at constant applied bias of 0.15 V vs. Fc/Fc⁺ (Supplementary Fig. 5), using TBABr as the electrolyte and Br⁻ source (mediator). As shown in Fig. 2c, after 2 h of PEC reaction, about 80% of the substrate **1** was converted, and the corresponding selectivity and FE for epoxide **2** formation were >99% and 82 ± 4%, respectively. Within 4 h of reaction, the substrate was completely consumed, and the final yield of **2** was 97 ± 1%. The monochromatic incident photon-to-electron conversion

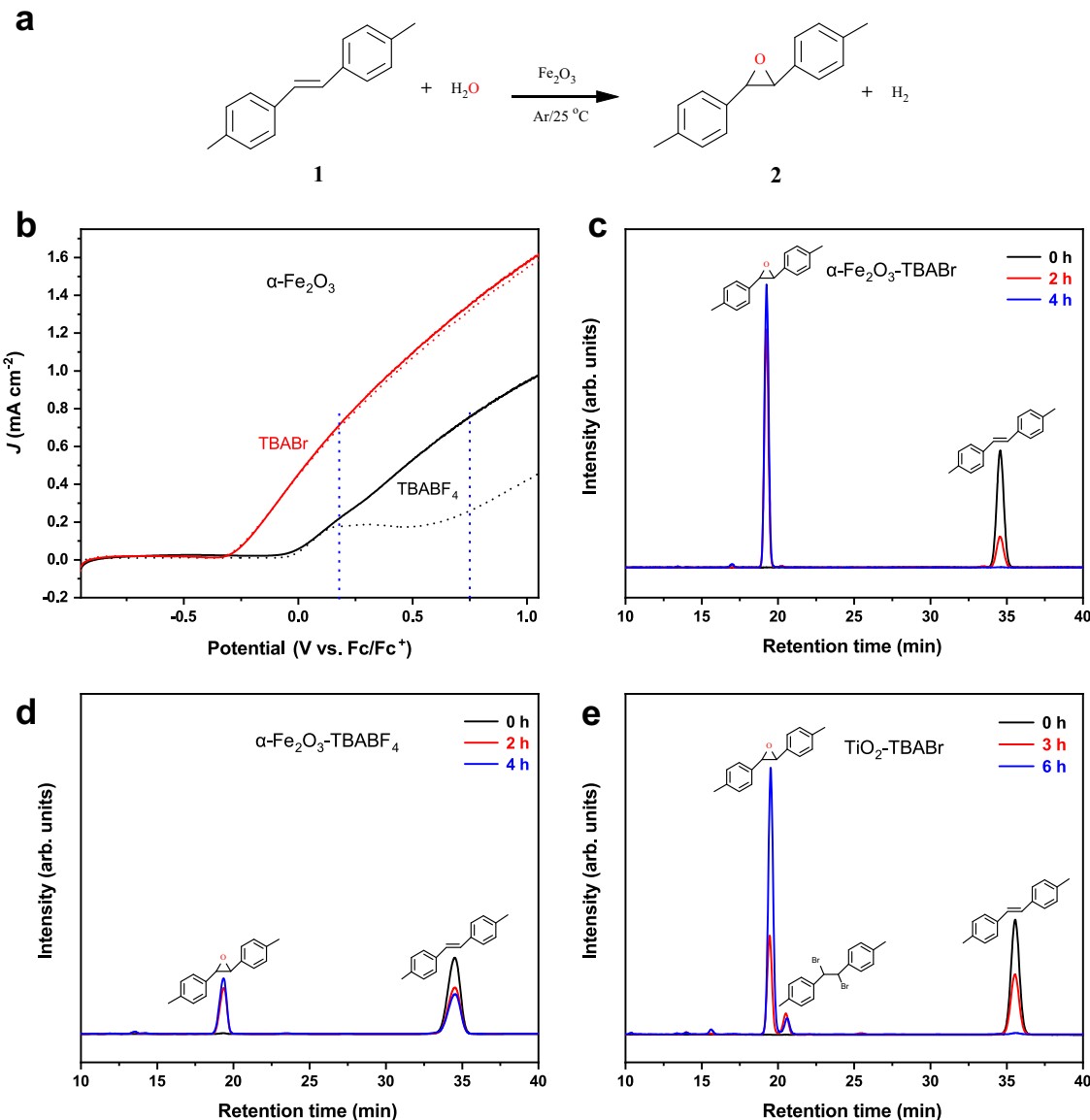

**Fig. 2 | Different PEC behaviors and the corresponding liquid chromatogram of epoxide products with different electrolytes and photoanodes. a** Reaction equation of Br-mediated epoxidation for substrate **1**. **b** $J-V$ scan of α-Fe$_2$O$_3$ under AM 1.5 G illumination measured in 100 mM TBABF$_4$ (black lines) and TBABr (red lines) solution (CH$_3$CN with 5% H$_2$O) under an Ar atmosphere without (dash) and with (solid) 10 mM **1**. Scan rate 0.05 V s$^{-1}$. The dotted vertical lines indicate the potentials (0.15 and 0.75 V vs. Fc/Fc$^+$) of photoelectrolysis experiments. **c** The HPLC spectra obtained at different PEC oxidation time of 10 mM **1** with 100 mM TBABr at 0.15 V vs. Fc/Fc$^+$ on α-Fe$_2$O$_3$. **d** The HPLC spectra obtained at different PEC oxidation time of 10 mM **1** with 100 mM TBABF$_4$ at 0.75 V vs. Fc/Fc$^+$ on α-Fe$_2$O$_3$. **e** The HPLC spectra obtained at different PEC oxidation time of 10 mM **1** with 100 mM TBABr at −0.05 V vs. Fc/Fc$^+$ on TiO$_2$.

efficiency (IPCE) experiments show that the maximum IPCE is ~21% at 0.55 V vs. Fc/Fc$^+$ at wavelength of 400 nm (Supplementary Fig. 6). To our knowledge, these performances are far beyond the most reported values during electrochemical and PEC epoxidation reactions (Supplementary Table 1). For the cathode, water was reduced to the hydrogen with a FE value of 90% (Supplementary Fig. 7). When the electrolyte was substituted by TBABF$_4$, in line with the LSV results (Fig. 2b), little photocurrent was observed at the applied bias of 0.15 V vs. Fc/Fc$^+$. By increasing the bias to 0.75 V vs. Fc/Fc$^+$, the photocurrent in the presence of **1** became comparable to that in TBABr system at 0.15 V vs. Fc/Fc$^+$. Even at this high bias, only 47 ± 7% of **1** was converted after 2 h of PEC reaction (Fig. 2d). At this point, the selectivity and FE for the epoxide products were only 43 ± 5% and 41 ± 3%, respectively. The final yield for **2** was only 23 ± 3% after 4 h of reaction, much lower than that in the TBABr system. When TBABr was replaced by NaBr (Supplementary Fig. 8), an excellent selectivity (>95%) of epoxide was

also achieved. However, when the Cl$^-$ and I$^-$ salts of TBA$^+$ (TBACl and TBAI) were used, the epoxidation activity had dramatically decreased under the same experimental conditions (Supplementary Figs. 9–10), indicative of the unique role of Br$^-$ in the epoxidation on α-Fe$_2$O$_3$. In addition, by increasing the ratio of Br$^-$ but keeping the total concentration of TBA$^+$ constant, the selectivity and FE values of epoxide **2** gradually increased (Supplementary Table 2). The higher yield and FE of the epoxide product in the presence of Br$^-$ confirm that bromide can serve as a mediator to promote the epoxidation reaction of alkenes.

The influences of the applied bias and pH on the Br-mediated epoxidation reaction were further examined. As shown in Supplementary Tables 3, the high selectivity values for the product **2** were obtained when the applied bias was in the range of 0.15 V-1.05 V vs. Fc/Fc$^+$, and the FE values exhibited an only slight decrease with the increasing bias. The effect of the acidic/basic properties for the solution on the Br-mediated epoxidation reaction was investigated by

changing the pH values of the added water (5 vol%). As shown in Supplementary Table 4, throughout the tested pH values of the added water from 3.0 to 11.0, the selectivity for the epoxide was quite excellent, while the FE values tended to have maximum at pH = 6.4. These experimental results demonstrate that the bromine-mediated epoxidation on α-Fe$_2$O$_3$ behaves excellent tolerance for the acid and the base. In previously reported Br-mediated electrochemical epoxidation on Pt electrode, the changes of pH values would greatly affect the distribution of the products[18,19]. The formations of dibromo product and bromohydrin are favored under acidic conditions, whereas diol product prefers to be formed under alkaline conditions. The different basic/acid effects on the α-Fe$_2$O$_3$ photoanode and the Pt anode suggest the distinct pathway for the Br-mediated epoxidation.

To investigate the unique behavior of α-Fe$_2$O$_3$ photoanodes for Br-mediated epoxidation, we compared the PEC oxidation of **1** on α-Fe$_2$O$_3$ and TiO$_2$, of which the latter is a typical single-electron oxidation photocatalyst. After photoelectrolysis of 3 h, 51 ± 4% of **1** was oxidized on the TiO$_2$ photoanode, and the FE value for epoxidation was only 39 ± 7%, which is also twice lower than that on α-Fe$_2$O$_3$ (82 ± 4%). The selectivity for epoxide **2** on TiO$_2$ (77 ± 8%) was also significantly lower than that on α-Fe$_2$O$_3$ (>99%). In addition, dibromo addition product of **1**, besides the epoxides, was evidently observed on TiO$_2$ (Fig. 2e), but was undetectable at all on α-Fe$_2$O$_3$. Moreover, the PEC performance for epoxidation on the α-Fe$_2$O$_3$ photoanode remained nearly unchanged after five repeated experiments (Supplementary Fig. 11). Also, after suffering from PEC reaction, the crystalline phase, the light absorption, the morphography and surface component of the α-Fe$_2$O$_3$ have not been changed obviously (Supplementary Figs. 2–3), indicative of stability of α-Fe$_2$O$_3$ in the PEC Br-mediated epoxidation reaction. On the contrary, the conversion, selectivity and FE for this reaction on the TiO$_2$ photoanode decayed much even in the second repeated experiments (Supplementary Fig. 11). In addition, some organic by-products are detected on the surface of TiO$_2$ after PEC reaction (Supplementary Figs. 12–13), which may be responsible for the decay in its performance. The bromine-mediated electrochemical epoxidation of **1** was further carried out on the Pt anode, which has been widely used in the earlier studies[17–19]. The poor performance (66% selectivity, 10% FE and 31% conversion at 5 h of electrolysis) was obtained under our experimental conditions, and the dibromo product was also detected (Supplementary Fig. 14). The FE value on α-Fe$_2$O$_3$ is eight times higher than that of Pt.

## Scope of the alkene substrates

The epoxidation of a broad range of aromatic and aliphatic alkenes was examined to explore scope of this PEC bromine-mediated epoxidation strategy on α-Fe$_2$O$_3$. As shown in Table 1 (Supplementary Figs. 15–26), *trans*-stilbene substrates with various electron-rich and electron-deficient substituents, including methyl (**2–6**), cyano (**7**), methoxy (**8, 9**) and halogen (**10–12**), could be efficiently oxygenated to the corresponding epoxides. It is observed that the *para*-methyl-sub-stitution of *trans*-stilbene (compare the precursors of **2, 3, 4, 5, 6**) significantly increased the FE value for their Br-mediated epoxidation. Moreover, with the increase of *para*-methyl groups (**4** with none *para*-methyl group, **3** with one *para*-methyl group and **2** two *para*-methyl groups), their FE values gradually increased (FE values: 58 ± 3% for **4**, 69 ± 3% for **3**, 82 ± 4% for **2**), indicative of that these substituted methyl groups can enhance the reaction activity between the Br-mediator and the C = C bond by enhanced electron cloud density on the C = C bond. For other *para*-groups substituted *trans*-stilbene, they were also efficiently oxygenated to the corresponding epoxides (**7–8, 10–11**) with satisfactory selectivity (90 to 98%) and FE values (50 to 65%), exhibiting a well compatibility for different electron-rich and electron-deficient groups. When one of the phenyl groups of **1** was replaced by a pyridine group (4-stilbazole, the precursor of **13**), the selectivity and FE were >99% and 53 ± 2%, respectively. 3-chloro-3′-methoxyl and

4,4′-dibromo- substituted *trans*-stilbenes exhibited excellent selectivity (**9**, 97 ± 3% and **12**, 98 ± 2%), but their FE values for epoxidation were somewhat low (**9**, 23 ± 2% and **12**, 27 ± 2%).

The epoxidation of styrene substrates (Table 1, Supplementary Figs. 27–36) with different *para*-groups such as halogen (**15–17**), -CN (**18**), alkyl (**19–20**) and phenyl (**21**) on α-Fe$_2$O$_3$ were also compatible and effective. The selectivity for the substrate with strong electron-deficient substitute (-CF$_3$ **22**) showed a slight decrease (81 ± 6%). For 2-vinyl-naphthalene, they can be oxidized to the corresponding epoxides **23** with the high selectivity and moderate FE value. For cyclooctene, the product **24** exhibited satisfactory selectivity (75 ± 4%) and FE (41 ± 1%) (Supplementary Table 5, Supplementary Fig. 37), of which both are much higher than those in the absence of Br-mediator (selectivity only 8.7 ± 0.5% and FE 2.7 ± 0.2% with TBABF$_4$ as electrolyte[14]). For other aliphatic alkenes (Supplementary Table 5, **25–29**, Supplementary Figs. 38–42) such as methyl-substituted cyclohexene, the quite good selectivity for the epoxidation reaction was observed. It should be noted that, relative to the aromatic alkenes, the selectivity and FE for the epoxidation of aliphatic alkenes is not very efficient, which may stem from the relative inertness of C = C bond of aliphatic alkenes toward epoxidation. Moreover, several by-products, including ketones, bromine-substituted products (as shown by GC-MS analysis, Supplementary Fig. 43) are detected. Nevertheless, the Br-mediated PEC epoxidation of aliphatic alkenes is still much higher than those on the bare α-Fe$_2$O$_3$. Moreover, the performance of the Br-mediated epoxidation of aliphatic alkenes is better than or comparable to that on MnO$_x$ and RuO$_2$ anodes or in the Br-mediated EC and PEC systems[3,33]. The results further confirmed that the indirect Br-mediated epoxidation showed the better catalytic activity and reaction compatibility.

## Different electrochemical performance and potential active bromine species

During the Br-mediated oxidation of alkenes, the key step is the oxidation of Br$^-$ on α-Fe$_2$O$_3$ photoanodes to form certain active species that can oxidize efficiently the alkene to its epoxide. The oxidation of Br$^-$ may undergo two possible pathways: OAT pathway (forming BrO$^-$ species, Eq. (1) and electron transfer pathway (leading to the bromine radical and then Br$_2$ molecule, Eq. (2). To investigate oxidation pathway of Br$^-$ on α-Fe$_2$O$_3$, we compared the PEC oxidation behavior of Br$^-$ on α-Fe$_2$O$_3$ and TiO$_2$ photoanodes and the electrochemical oxidation on the Pt electrode (Supplementary Fig. 44). As mentioned above, the onset potential in the electrolyte of TBABr is much lower (by $\Delta E_{onset}$ = 300 mV) than that in the TBABF$_4$ on α-Fe$_2$O$_3$ (Figs. 2b and 3a). However, the oxidation onset potentials in the Br$^-$ system on Pt ($\Delta E_{onset}$ = 90 mV, Fig. 3b) and on TiO$_2$ ($\Delta E_{onset}$ = 120 mV, Fig. 3c) exhibited only slightly negative shift, relative to those in the BF$_4^-$ system. The much larger onset potential shift on α-Fe$_2$O$_3$ than that on other electrodes suggests the distinct oxidation mechanism of Br$^-$ on α-Fe$_2$O$_3$. The redox potential for oxidation of Br$^-$ to Br$_2$ ($E_0$ = 1.07 V vs. NHE, Eq. 2) is just lower a bit than that for water oxidation ($E_0$ = 1.23 V vs. NHE, Eq. 3). The small negative potential shift on Pt and TiO$_2$ anodes suggests that the oxidation of Br$^-$ on these electrodes may proceed dominantly through an electron transfer pathway to Br$_2$. Actually, it has been reported that the Br-mediated reaction on Pt anodes occurs through a Br$^-$/Br$_2$ pathway, in which Br$^-$ is oxidized to bromine radical by a single-electron process and then form Br$_2$[18–20,22,34]. TiO$_2$ have a high valence band potential ($E_0^{vb}$ = 3.1 V vs. NHE)[35,36]. Its photoinduced valence band hole can facilely oxidize Br$^-$ to Br radical, and H$_2$O to OH radical. On the other hand, the redox potential for Br$^-$/BrO$^-$ ($E_0$ = 0.76 V vs. NHE, Eq. 1) is much lower than that of water oxidation and Br$^-$/Br$_2$. The large potential shift on α-Fe$_2$O$_3$ implies that the OAT (Br$^-$/BrO$^-$) may be dominant pathway on α-Fe$_2$O$_3$. In our previous study, we found that α-Fe$_2$O$_3$ can serve as an efficient OAT catalyst to realize the oxygenation of inorganic and organic substrates[14]. It is reasonable that the PEC oxidation of Br$^-$

**Table 1 | Scope of the substituted stilbene and styrene substrates for the PEC bromine-mediated alkene epoxidation[a]**

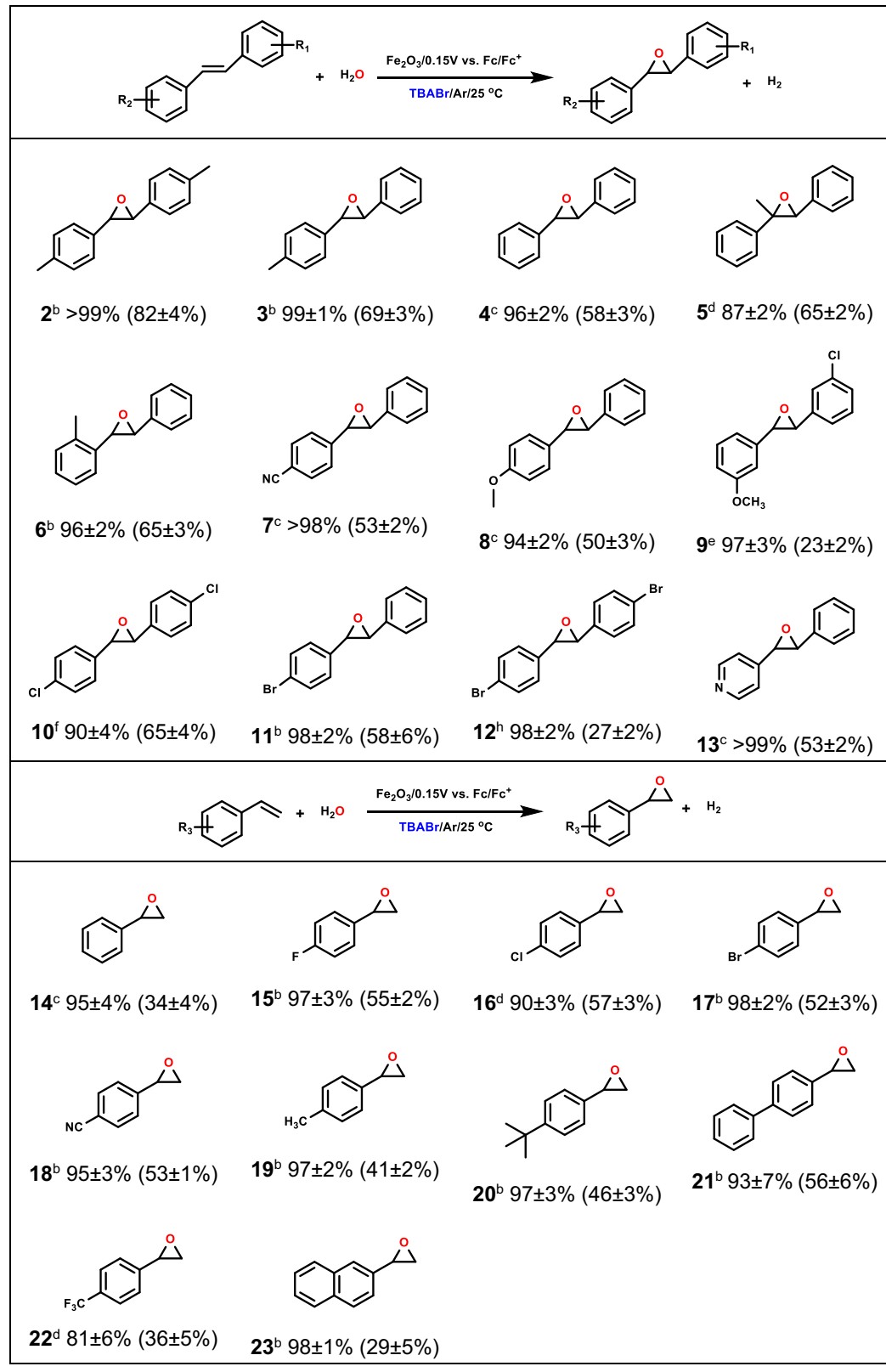

[a]Reaction conditions: substrates (10 mM) in $CH_3CN$ (5% $H_2O$) in Ar atmosphere at room temperature with 0.15 V vs. Fc/Fc+ applied bias. Selectivity and FE values (in brackets) were determined by HPLC and [1]H NMR (Supplementary Figs. 15–26 and 27–36). All the error bars are defined in the table (s.d., $n = 3$ independent experiments) together with a measure of the mean.
[b]2.0 h of reaction.
[c]2.5 h of reaction.
[d]1.5 h of reaction.
[e]5.0 h of reaction.
[f]5.0 mM of **10** with 1.0 h of reaction.
[g]1.0 mM of **12** with 0.5 h of reaction.

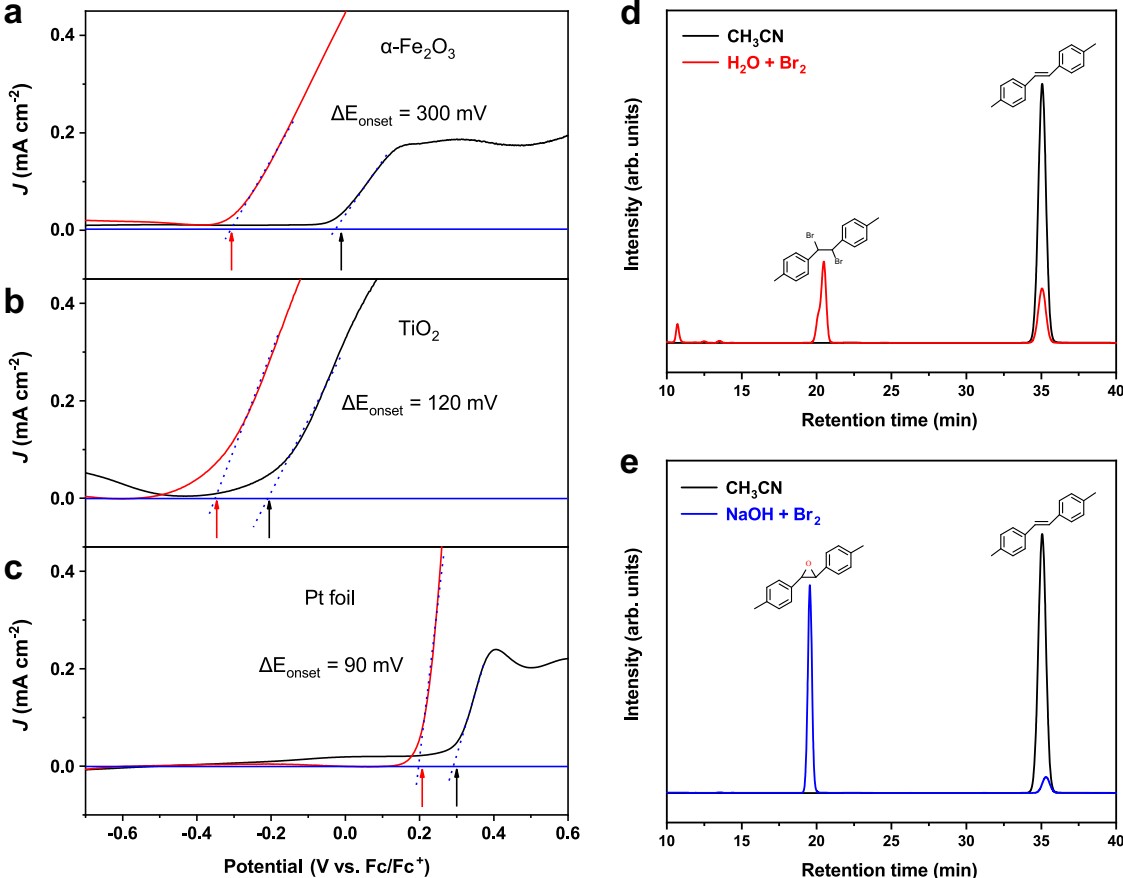

**Fig. 3 | J–V scan and the product distribution.** Comparison on J–V scan of (**a**) α-Fe$_2$O$_3$, (**b**) TiO$_2$ under AM 1.5 G illumination, and (**c**) Pt foil electrode measured in 100 mM TBABF$_4$ (black lines) and TBABr (red lines) solution (CH$_3$CN with 5% H$_2$O) under an Ar atmosphere. The arrows mark the onset potentials; The product distribution for oxidation of **1** after adding (**d**) Br$_2$ and (**e**) NaOH-Br$_2$ aqueous solution (5% v:v) into the CH$_3$CN solution of **1** (ice bath).

occur through OAT pathway, which results in the formation of BrO$^-$ species, and electron transfer pathway is minimized.

To make clear the identification of active species during the Br-mediated epoxidation, potential active bromine species were generated in in-situ manner under our electrolytic conditions, and the product distribution for oxidation of alkene **1** by these species was examined. The bromine radical was produced through the single-electron oxidation of Br$^-$, by photosensitized reduction of Na$_2$S$_2$O$_8$ using [Ru(bpy)$_3$]Cl$_2$ as a photosensitizer[37]. As shown in Supplementary Fig. 45, only dibromo addition product was observed, but little epoxide was obtained in this dye-photosensitized system, excluding that bromine radical is the main active intermediate species in the Br-mediated epoxidation system on α-Fe$_2$O$_3$. We further examined the activity of mixture of H$_2$O and Br$_2$ for the oxidation of **1** at low-temperature (ice bath, details in Methods), and its dibromo addition product was found to be dominant (Fig. 3d). Under these conditions, besides Br$_2$, Br radical is also formed[19,20]. We propose that the dibromo product is attributed to Br$_2$ and Br radical. Such a proposal is also supported by the appearance of dibromo addition product on the Pt and TiO$_2$ anodes (Supplementary Fig. 14 and Fig. 2e). Further, BrO$^-$ species were generated by fiercely stirring of 1 M NaOH and Br$_2$ at low temperature (ice bath)[38]. As shown in Fig. 3e, like that on α-Fe$_2$O$_3$, the mixture of substrate **1** and BrO$^-$ species produced exclusively epoxide **2**. In addition, the epoxidation in the presence of H$_2$O$_2$ shows poor activity on α-Fe$_2$O$_3$ (Supplementary Fig. 46), excluding the possibility that H$_2$O$_2$ geneated by water oxidation can act as the oxidant for the epoxidation of alkenes. These results verify that BrO$^-$ species can serve as the actual intermediate active species in the Br-mediated epoxidation system.

## Proposed Br$^-$/BrO$^-$-mediated epoxidation mechanism

H$_2$$^{18}$O isotopic labelling experiments were performed to identify the origination of oxygen atom. Based on mass spectra in Supplementary Fig. 47, the prominent molecular ion peaks of product **2** appeared at a mass-to-charge ratio (m/z) of 226.0 and 224.0, when H$_2$$^{18}$O and H$_2$$^{16}$O were used in an Ar atmosphere, respectively. The shifts of two m/z units indicate that water serves as the only oxygen atom source in the Br-mediated epoxidation process. Based on the above analysis, the bromine mediated epoxidation mechanism was proposed in Fig. 4. In our previous work[14], the accumulation of two-hole on hematite had been confirmed to behave the better OAT activity than that of single hole oxidation. In the bromine mediated epoxidation system, two-hole oxidation should be also suitable. Therefore, it is believed that surface-trapped states are generated with a step of two-hole transfer to surface-coordinated hydroxyl groups and a concomitant depro-tonation (step *a* in Fig. 4). Then, surface-trapped holes (Fe$^{IV}$ = O species) can oxide the bromide ion to produce the intermediate (BrO$^-$ species) via an OAT pathway (step *b* in Fig. 4), in which two electrons are transferred in one kinetic step, and no radical is involved. For step *c*, the BrO$^-$ species can directly transfer oxygen atom to the alkene, in which only epoxidation product was detected without any bromohydrin species or dibromo addition products. By contrast, TiO$_2$ photoanodes, Pt anodes and dye-assistant system produced dibromo products through the Br$^-$/Br$_2$ pathway[19,20,34]. Based on the epoxidation reaction kinetics of various para-substituent *Trans*-stilbene substrates, the Hammett linear free-energy relationship was calculated and showed a negative slope of −0.20 (Supplementary Fig. 48), which confirms an electrophilic

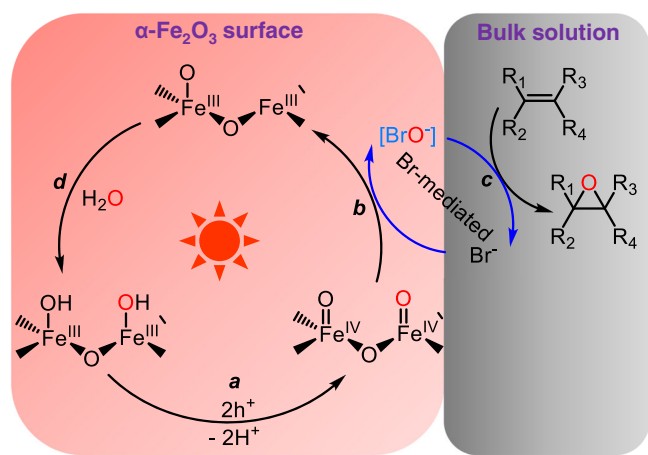

**Fig. 4 | Schematic illustrations for epoxidation of alkenes by a Br⁻/BrO⁻-mediated pathway.** In the process of Br-mediation epoxidation, the BrO⁻ species are generated via an OAT pathway from Fe^{IV} = O species on the surface of α-Fe₂O₃, and then transfer the oxygen atom to the alkene in the bulk solution.

attack of BrO⁻ on the double bond of alkenes[39,40]. Moreover, no obvious reversible redox peak was detected (Supplementary Fig. 49), suggesting that benzylic cations or radical cations are not involved in the epoxidation process. Notably, the photocurrent in the Br-mediated epoxidation system is nearly unchanged by addition of the alkene substrate (Fig. 2b). If the reaction between BrO⁻ and alkene takes place on the surface, the regeneration of Br⁻ by alkene would increase the local concentration of Br⁻ on the surface of the electrode, after adding the alkene substrate. As a result, the current of Br⁻ oxidation should be enhanced. However, no current enhancement by adding alkene demonstrates that Br-mediated oxidation reaction of alkene takes place in the bulk solution, rather than on the surface of photoanode. The step $d$ is a process of water adsorption and dissociation at reaction sites to regenerate the surface hydroxyl groups and terminate the catalytic cycle, as confirmed by $H_2^{18}O$ isotopic labelling experiments. In Br-mediated epoxidation process, therefore, bromide acts as an OAT mediator to conduct efficient epoxidation reaction, and BrO⁻ species are the active intermediate with a direct OAT pathway from Fe^{IV} = O species to the Br⁻ species. It is also notable that some $BrO_3^-$ were detected by ion chromatograph (IC), accounting for a FE of 13% (Supplementary Fig. 50), which indicates that the overoxidation of Br⁻ may occur during Br-mediated epoxidation process. In addition, no O₂ evolution was observed (Supplementary Fig. 7). These results imply that the loss in FE is mainly attributed to the overoxidation of Br⁻, rather than to the water oxidation.

In summary, we have developed a Br⁻/BrO⁻-mediated system on α-Fe₂O₃ instead of the direct electrochemical and PEC systems to achieve more efficient PEC epoxidation activity of alkenes under mild conditions. The Br⁻/BrO⁻-mediated epoxidation system exhibits satisfactory selectivity, FE and substrate universality with water as the only oxygen source and hydrogen as the cathodic product. By comparing with different electrochemical behaviors and product distribution in different systems, we proposed that BrO⁻ species were the reaction active species via an OAT pathway on α-Fe₂O₃ and subsequently transferred the oxygen atom to the alkenes. We believe that this strategy of Br⁻/BrO⁻-mediated efficient epoxidation will provide a versatile applied potential for value-added and fine chemical products.

## Methods

### General information
All chemicals were purchased and used without further purification, except noted.

### Photoanode preparation
A two-step method (hydrothermal and annealing) was used to prepare the photoanodes in this study based on previous reports[14,41]. For the synthesis of α-Fe₂O₃ photoanodes, a 100 mL aqueous solution containing anhydrous ferric chloride (2.43 g) and sodium nitrate (0.85 g) was vigorously stirred for 30 min. A 10 mL aliquot of the solution was then added to a 20 mL reaction kettle containing a piece of clean FTO glass that had been treated with ultrasonic cleaning in acetone, ethanol, and water for 30 min each. The FTO glass was coated with FeOOH using a hydrothermal method (95 °C for 4 h), followed by sintering at 550 °C for 2 h and annealing at 750 °C for 15 min. TiO₂ photoanodes were prepared according to a previously reported method. In brief, titanium n-butoxide (0.8 ml) was mixed with a solution of 35–37% HCl and deionized water (30 ml each) and stirred for 30 min. The resulting solution and FTO glass were then transferred to a Teflon reactor for hydrothermal synthesis at 150 °C for 4 h. The obtained precursor was washed with deionized water and annealed at 450 °C for 1 h.

### Electrochemical study
A single compartment cell was used for electrochemical testing, in which a standard three-electrodes system was employed (α-Fe₂O₃ as the working electrode, Ag/AgCl as the reference electrode, Pt wire as the counter electrode). The CHI-760E electrochemical workstation and a 500 W xenon lamp (100 mW cm⁻², Microsolar300, Beijing Perfectlight) with an AM 1.5 G filter were used to conduct the PEC experiments, which included linear sweep voltammetry and photoelectrolysis experiments. The potentials (versus Ag/AgCl) were calibrated using a 5.0 mM ferrocene/ferrocenium redox couple, as shown in Supplementary Fig. 51 (E(vs. Fc/Fc⁺) = E(vs. Ag/AgCl) − 0.45 V).

### Product analysis
All aromatic alkenes were analyzed and quantified with a high-performance liquid chromatography (HPLC, Agilent), in which a Dikma Diamond C-18(2) column (250 × 4.6 mm, 5 µm film thickness) was employed. For the detection of these substrates, different proportions of water and acetonitrile were served as the mobile phase, in which the flow rate and detection wavelength were also adjusted to better detect the different alkenes. The detailed parameters were set as follows: **2**, **8** and **9** ($V_{acetonitrile}/V_{water}$ = 75/25, 0.3 ml min⁻¹ and 230 nm); **3**, **6**, **7**, **10** and **11** ($V_{acetonitrile}/V_{water}$ = 70/30, 0.3 ml min⁻¹ and 240 nm); **4** ($V_{acetonitrile}/V_{water}$ = 60/40, 0.2 ml min⁻¹ and 240 nm); **5** and **21** ($V_{acetonitrile}/V_{water}$ = 75/25, 0.3 ml min⁻¹, 240 nm for **5** and 254 nm for **21**); **12** ($V_{acetonitrile}/V_{water}$ = 85/15, 0.3 ml min⁻¹ and 254 nm); **14** ($V_{acetonitrile}/V_{water}$ = 70/30, 0.2 ml min⁻¹ and 210 nm); **13**, **15**, **16**, **18**, **20** and **22** ($V_{acetonitrile}/V_{water}$ = 65/35, 0.2 ml min⁻¹, **15** and **16** for 210 nm, **13** and **18** for 240 nm, **20** and **22** for 230 nm); **17**, **19** and **23** ($V_{acetonitrile}/V_{water}$ = 70/30, 0.2 ml min⁻¹ and 230 nm).

The detection of aliphatic alkenes was used by gas chromatography (GC, Agilent GC7890B) with FID detector and a DB-VRX column (20 m × 180 µm × 1 µm). For the detection of **24** and **29**, the injection and detector temperatures were 200 and 230 °C, respectively. The GC conditions were used as follows: **24**, 50 °C (2 min), 10 °C min⁻¹, 120 °C (2 min), 10 °C min⁻¹ and 185 °C (0 min); **29**, 50 °C (5 min), 20 °C min⁻¹, 110 °C (0 min), 10 °C min⁻¹ and 180 °C (0 min). For 25–28, the injection and detector temperatures were 200 and 250 °C, respectively. The GC conditions were used as follows: **24**–**26** and **28**, 50 °C (4 min), 20 °C min⁻¹, 110 °C, 20 °C min⁻¹ and 210 °C; **27**, 50 °C (4 min), 20 °C min⁻¹, 110 °C (2 min), 40 °C min⁻¹ and 210 °C (2.5 min).

The GC-MS system was used to detect the isotope compositions of the products **2** during the epoxidation of **1** in $H_2^{18}O$ isotopic labelling experiments, in which the GC conditions were shown as follows: HP-5MS column (30 m × 250 µm × 0.25 µm), 50 °C (3 min), 20 °C min⁻¹, 170 °C (0 min), 10 °C min⁻¹, 280 °C (2 min). The corresponding injection and detector temperatures were 180 °C and 300 °C, respectively.

The $^1$H NMR spectra of the corresponding epoxides (approximated 10.0 mM) generated during PEC epoxidation were measured using a 400 MHz Bruker instrument. The chemical shifts were calibrated by using residual solvent peaks ($CD_3CN$) as reference. To quantify the epoxides, an internal standard of 1,3,5-trimethoxybenzene was added, which is stable in the system.

## The experiments of different potential active bromine species with substrate

For $H_2O + Br_2$ species, $Br_2$ molecule (0.5 M, 256 μL) was added into 10 mL $H_2O$ and the mixture solution was stirring for 30 min under ice-bath condition. The obtained solution (1 mL) was transferred into 19 mL $CH_3CN$ with stirring under ice-bath condition to obtain 25 mM $Br_2 + H_2O$ solution ($V_{water}/V_{acetonitrile} = 1/19$). Finally, 1 mL 25 mM $Br_2 + H_2O$ and 1 mL 10 mM substrate **1** solution were mixed and stirred thoroughly, and the mixed solution was tested by HPLC.

For NaBrO species, $Br_2$ molecule (256 μL) was added into 10 mL 1 M NaOH solution and the mixture solution was stirring for 30 min under ice-bath condition. The obtained solution (1 mL) was transferred into 19 mL $CH_3CN$ with stirring under ice-bath condition to obtain 25 mM NaBrO solution ($V_{water}/V_{acetonitrile} = 1/19$). Finally, 1 mL 25 mM NaBrO and 1 mL 10 mM substrate **1** solution were mixed and stirred thoroughly, and the mixed solution was tested by HPLC.

For Br radical, 1 mM [Ru(bpy)$_3$]Cl$_2$, 10 mM substrate **1**, 100 mM TBABr and 20 mM $Na_2S_2O_8$ were add into 5 mL $CH_3CN$ with 5% $H_2O$. The solution was irradiated under simulated sunlight (AM 1.5 G), and the solution was tested by HPLC.

## Characterization

The X-ray diffraction (XRD) spectra were tested on an X-ray diffractometer (Empyrean, PANalytical) by using Cu-Kα radiation (scan rate of 0.05° 2θ s$^{-1}$). UV-vis spectra were detected on a UV-vis spectrophotometer (Hitachi U-3900). The measurement of X-ray photoelectron spectroscopy (XPS, 300 W Al-Kα radiation) was performed on an ESCALAB 250Xi spectrometer, whose binding energy was calibrated with respect to the C 1s level 284.8 eV of contaminated carbon. Scanning electron microscopy (SEM) and transmission electron microscopy (TEM) images were conducted with an SU8010 (Hitachi, Japan) and HT770 (Hitachi, Japan), respectively. Incident photon to current conversion efficiency (IPCE) was collected under light from a 500 W xenon lamp through a monochromator.

## Data availability

The authors declare that the data supporting the findings of this study are available within the paper and its supplementary information files.

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

## Acknowledgements

This work was supported by the Strategic Priority Research Program of Chinese Academy of Sciences (No. XDB36000000), the National Natural Science Foundation of China (No. 22136005, 22106164, 22188102, 21827809), the National Key R&D Program of China (No. 2018YFA0209302, 2020YFA0710303), and the China Postdoctoral Foundation (No. 2021M703269).

## Author contributions

Y. Zhao and C.C. conceived and designed the experiments. Y. Zhao performed most of the experiments. M.D. prepared and characterized α-$Fe_2O_3$ and $TiO_2$ anodes. C.D. and J.Y. contributed to the tests of some substrates. Y. Zhao, M.D., C.D., J.Y., S.Y., Y. Zhang, H.S., Y.L., C.C. and J.Z. analyzed the results and reviewed the paper. Y. Zhao and C.C. wrote the paper, with input from the other authors.

## Competing interests

The authors declare no competing interests.
