## [Peer review file · Nature Communications]

REVIEWER COMMENTS

Reviewer #1 (Remarks to the Author):

The manuscript by Chen et al. reported a high efficiently photoelectrochemical epoxidation research through Br-/BrO- mediated route with oxygen atom transfer pathway on α -Fe₂O₃ anode. In comparison with the reported methods, the PEC Br-/BrO- mediated epoxidation system shows clear advantages for satisfied selectivity and Faradaic efficiency for a wide range of alkenes. The authors have verified the epoxidation process mediated by Br-/BrO- route with an oxygen atom transfer pathway, and conducted broad substrates to illustrate the universality of the methods. The manuscript is very interesting and worth to studying for the efficient synthesis of epoxides, however, it still needs to solve some nonnegligible problems before publishing in this journal.

- 1). The specific structure and elemental state of α -Fe₂O₃ are necessary to provide for a heterogeneous electrode, even if the same material has been reported in your previous reports, such as XRD, XPS, and TEM, etc., which contribute to understanding deeply the changes of structure and metal valence before and after reaction. Additionally, the same characterizations of the used electrode are needed.
- 2). In this manuscript, above 80% of Faradaic efficiency for epoxidation was achieved on α -Fe₂O₃ anode, however, what reactions happen for the rest electrons? Hydrogen peroxide is a nonnegligible product during the oxidation process of water molecules, and is also the common oxidant in the epoxidation of alkene. And oxygen gas evolved on the anode was reduced on the Pt cathode to oxygen species, and then participating in the epoxidation reaction. Whether the author excludes the possibilities in the PEC system?
- 3). The authors should add the epoxidation-based incident photon to current efficiencies in the PEC system.
- 4). The TBABr seems act as an pre-mediator for the epoxidation. The possibility of other borides or BrOx-containing compounds to initiate the same reaction should be carefully tested.
- 5). Similarly, the possibility of I-/IOx-, or Cl-/ClO- is also to be tested to further extend the mediator scope. If NOT, the unique role of Br-/BrO- should be well discussed.

Reviewer #2 (Remarks to the Author):

In this work, the authors developed an approach to oxidize alkenes to epoxides using Br-/BrO- as the mediator for electron transfer. This report is well-written and the authors provided sufficient evidence to show that alkene oxidation is by BrO- rather than Br₂. Isotope experiments further demonstrated

that oxygen is from water oxidation. However, a significant drawback of this paper is that this alkene epoxidation only can reach a good faradaic efficiency when alkene contains phenyl functional groups. This severely limits the application of this approach and should be mentioned and explained in the manuscript. Another problem that I found is that the authors cited ref 5 as an example of indirect electrochemical epoxidations. However, in the original paper, it was assumed that the reaction was through direct epoxidation. Thus, this should be corrected. Overall, this is a good paper, and I support its publication on Nat. Commun.

Reviewer #3 (Remarks to the Author):

The authors report a photoelectrochemical route through which epoxidation is mediated by Br, driving ultimate oxygen atom transfer from water to olefins.

The authors claim in the introduction that electrochemical routes suffer from poor selectivity (line 38-41). I don't think this is substantiated by the literature. For instance the following work shows high FE for epoxidation:

<https://www.science.org/doi/full/10.1126/science.aaz8459>

The work achieves 70% FE for ethylene and propylene. This is one of several examples, others are cited in the paper. The authors should state these results clearly so that the state of the field can be gauged accurately by the reader.

Since these results demonstrate that electrochemically a halogen mediator can be used, it is not clear what problem the introduction of a photon solves. Can the authors articulate specifically why introducing light is important?

Can the authors conduct an energy balance for pure electrochemical polarization vs the photoelectrochemical route they are conducting? This might highlight potential advantages of disadvantages of their route.

That being said, the implementation of a photoelectrochemical scheme is of fundamental interest and will inspire other work in the field.

Reviewer #4 (Remarks to the Author):

The authors present the photoelectrochemical (PEC) epoxidation of mostly aryl and diaryl substituted alkenes using alpha-Fe₂O₃ as photo anode and Pt as cathode and Br⁻ as redox mediator.

Overall, the synthetic results for aryl and diaryl substituted alkenes are very good in terms of selectivity and yield with a significant variation in faradaic efficiency, Table 1.

The authors claim that hypobromite, BrO⁻, is the oxidising species.

I believe this research deserves publication in an organic chemistry oriented journal such as the Journal of Organic Chemistry or Organic Letters but not in Nature Communications

1. While the synthetic results for aryl and diaryl substituted alkenes are very good, the results for aliphatic cyclic and acyclic alkenes are not with mediocre selectivity (the byproducts were not given) and poor faradaic efficiency.

2. The hypothesis that hypobromite is an interesting one, but it is not proven. It is always somewhat dangerous to carry out mechanistic studies on the most reactive substrate only. Such studies tend to hide other pathways, precisely because of their high reactivity.

(a) where does the epoxidation reaction take place? In solution or on the electrode?

(b) How is hypobromite stable at pH -3, as the results in Table S3 suggests when the pK_a of HOBr is 7.5.

3. The authors use "solar light" for the experiments. Which wavelengths are relevant?

What is the quantum yield?

4. The epoxidation appears to be the result of an electrophilic attack of BrO⁻ on the double bond but this should be demonstrated through use of kinetics and Hammett plots. See Chemical Science, 2020, 11, 11584 – 11591 as an example.

5. The reaction also occurs in the absence of Br⁻, Figure 2c albeit at somewhat higher potentials. How do these reactions proceed, how? at what selectivity? Has

6. In some of the GC plots it looks like there are two products that are poorly separated. Why is that?

7. The authors should also disclose the possibility of formation of benzylic cations or radical cations, followed by nucleophilic attack. CV's of some substrates would be a good start.

Summary - An organic chemistry oriented journal such as the Journal of Organic Chemistry or Organic Letters is more reasonable for this research as it stands. Nature Communications would require a significant amount of additional research.

Response to Reviewer's Comments

(Reviewer's comments in black font. Our response in blue font. *Revision in manuscript in yellow background.*)

Thanks to all reviewers for having given us valuable comments and suggestions on the manuscript of *NCOMMS-22-38942* submitted to the journal of ***Nature Communications***.

To Reviewer #1:

The manuscript by Chen et al. reported a high efficiently photoelectrochemical epoxidation research through Br⁻/BrO⁻ mediated route with oxygen atom transfer pathway on α -Fe₂O₃ anode. In comparison with the reported methods, the PEC Br⁻/BrO⁻ mediated epoxidation system shows clear advantages for satisfied selectivity and Faradaic efficiency for a wide range of alkenes. The authors have verified the epoxidation process mediated by Br⁻/BrO⁻ route with an oxygen atom transfer pathway, and conducted broad substrates to illustrate the universality of the methods. The manuscript is very interesting and worth to studying for the efficient synthesis of epoxides, however, it still needs to solve some nonnegligible problems before publishing in this journal.

Response: Thanks for the reviewer's precious comments and suggestions. We have revised our manuscript according to the reviewer's suggestions. Particularly, more characterizations for fresh and used photoanodes (α -Fe₂O₃ and TiO₂), including XRD, UV-vis, XPS, SEM and TEM, have been performed. The tests of incident photon to current efficiencies (IPCE) and other electrolytes (NaBr, NaBrO₃, TBACl and TBAI) were also conducted. The corresponding results and discussions have been supplied in the revised manuscript and supplementary information. The point-by-point responses are presented as follows.

1). The specific structure and elemental state of α -Fe₂O₃ are necessary to provide for a heterogeneous electrode, even if the same material has been reported in your previous reports, such as XRD, XPS, and TEM, etc., which contribute to understanding deeply the changes of structure and metal valence

before and after reaction. Additionally, the same characterizations of the used electrode are needed.

Response: As the reviewer suggested, we have provided detailed characterizations (XRD, UV-vis, SEM, TEM and XPS) of α -Fe₂O₃ and TiO₂ photoanodes, and compared their changes after the Br-mediated PEC reaction. As depicted in **Figs. R1-R2**, the characterizations demonstrate that the α -Fe₂O₃ photoanode consists of nanorods with the diameter of 50-100 nm in the crystalline phase of hematite, which shows a good visible-light response with band gap of \sim 2.1 eV. The corresponding valence states of Fe and O elements on α -Fe₂O₃ are +3 and -2, respectively. After suffering from PEC reaction, the crystalline phase (XRD), the light absorption (UV-vis), the morphology (SEM and TEM) and surface components (including valence states, XPS) have not been changed obviously, indicative of that α -Fe₂O₃ is stable in the PEC Br-mediated epoxidation reaction, which is consistent with the results of repeated experiments (**Fig. S11**).

For TiO₂, the XRD and UV-vis spectra suggest that the crystalline phase of TiO₂ is rutile with a band gap of \sim 3.1 eV (**Figs. R3a and R3b**). The SEM and TEM images (**Figs. R3c and R3e**) show that the TiO₂ photoanode consists of uniform nanorods with the diameter of around 100 nm. In XPS spectra (**Fig. R4**), the corresponding valence states of Ti and O elements on TiO₂ are +4 and -2, respectively. Different from α -Fe₂O₃ photoanodes, as shown in **Fig. R3**, TiO₂ anode is found to be changed significantly after the Br-mediated PEC reaction. Particularly, in the used TiO₂ photoanode, a new absorption in the wavelength range of 400 – 650 nm in the UV-vis spectra appears (**Fig. R3b**). More O1s and C1s peaks in the XPS spectra are observed (**Fig. R4**). The SEM image shows that the surface of anode seems to be covered by some amorphous species (**Fig. R3d**). All these results suggest that some organic by-products during PEC Br-mediated epoxidation process are formed and deposited on the surface of TiO₂, which is consistent with the poorer selectivity and the rapid decay in the PEC activity of TiO₂.

The above results and discussions have been added into revised manuscript (Lines 85-89, Page 3; Lines 160-165, Page 4) as “*The characterizations demonstrate that α -Fe₂O₃ photoanode consists of nanorods with the diameter of 50-100 nm in the crystalline phase of hematite, which shows a good visible-light response with band gap of ~ 2.1 eV. The corresponding valence states of Fe and O elementals on α -Fe₂O₃ are +3 and -2, respectively (see Figs. S2-S3 for more detailed description); Also, after suffering from PEC reaction, the crystalline phase, the light absorption, the morphology and surface component of the α -Fe₂O₃ have not been changed obviously (Figs. S2-S3), indicative of stability of α -Fe₂O₃ in the PEC Br-mediated epoxidation reaction; In addition, some organic by-products are detected on the surface of TiO₂ after PEC reaction (Figs. S12-S13), which may be responsible for the decay in its performance*”. More detailed descriptions are provided in the revised supplementary information (*under the Figs. S2-S3 and S12-S13*).

Fig. R1 The characterizations of fresh and used $\alpha\text{-Fe}_2\text{O}_3$ photoanodes. (a) XRD spectra; (b) UV-vis diffuse spectra; SEM images of fresh (c) and used (d) $\alpha\text{-Fe}_2\text{O}_3$; TEM images of fresh (e) and used (f) $\alpha\text{-Fe}_2\text{O}_3$.

Fig. R2 The XPS spectra of fresh and used α -Fe₂O₃ photoanodes. (a) XPS spectra of Fe 2p core level; (b) XPS spectra of O 1s core level; and (c) XPS spectra of C 1s core level.

Fig. R3 The characterizations of fresh and used TiO₂ photoanodes. (a) XRD spectra; (b) UV-vis diffuse spectra; SEM images of fresh (c) and used (d) TiO₂; TEM images of fresh (e) and used (f) TiO₂.

Fig. R4 The XPS spectra of fresh and used $\alpha\text{-Fe}_2\text{O}_3$ photoanodes. (a) XPS spectra of Ti 2p core level; (b) XPS spectra of O 1s core level; and (c) XPS spectra of C 1s core level.

2). In this manuscript, above 80% of Faradaic efficiency for epoxidation was achieved on $\alpha\text{-Fe}_2\text{O}_3$ anode, however, what reactions happen for the rest electrons?

Response: Due to high selectivity of epoxide (> 99%), the possibility of other oxidation pathways of the alkene substrates (e.g. to form diol, aldehyde) can be safely excluded. The oxidation of water to produce O_2 or/and overoxidation of bromide may account for the rest electrons in our system. To examine the contribution of these two reactions, the gas in the headspace after the PEC reaction was analyzed by gas chromatography (GC). No dioxygen (if formed, its retention time should be at ~ 1.6 min) was detected (**Fig. R5**) in GC spectra, which rules out that water oxidation to O_2 is a significant contribution to the rest currents. By ion chromatography (IC), some BrO_3^- ions were detected (**Fig. R6**), which accounts for a FE of 13%. After considering the formation of BrO_3^- , the total FE reaches 95%. Therefore, the overoxidation of Br^- to BrO_3^- ions is dominantly responsible for the lost FE.

In the revised version, we have added the above results and discussion in the revised manuscript (Lines 293-298, Page 7) as *“It is also notable that some BrO_3^- were detected by ion chromatograph (IC), accounting for a FE of 13% (Fig. S50), which indicates that the overoxidation of Br^- may occur during Br^- ”*

mediated epoxidation process. In addition, no O_2 evolution was observed (Fig. S7). These results imply that the loss in FE is mainly attributed to the overoxidation of Br^- , rather than to the water oxidation.” and revised supplementary information (Fig. S50).

Fig. R5 The GC spectra of the headspace gas of the PEC cell after 4 hours' photoelectrolysis.

Fig. R6 The IC spectra of reaction solution after 2 hours' photoelectrolysis.

3). Hydrogen peroxide is a nonnegligible product during the oxidation process of water molecules, and is also the common oxidant in the epoxidation of alkene. And oxygen gas evolved on the anode was reduced on the Pt cathode to oxygen species, and then participating in the epoxidation reaction. Whether the

author excludes the possibilities in the PEC system?

Response: According to the result of GC (**Fig. R5**), dioxygen is not formed during the Br-mediated epoxidation process, which means that water oxidation is much less competitive than bromide oxidation on $\alpha\text{-Fe}_2\text{O}_3$ photoanodes, consistent with LSV results in **Fig. 2a**. Thus, little hydrogen peroxide should be generated during the PEC reaction. As the reviewer pointed out, hydrogen peroxide (H_2O_2) can act as a common oxidant for the epoxidation of alkenes in the presence of catalysts (such as titanium silicalite-1, metal-based complex). To further exclude the possibility of participation of H_2O_2 in our system, we conducted the epoxidation experiment with 10 mM alkene and 10 mM H_2O_2 on $\alpha\text{-Fe}_2\text{O}_3$. After 4 hours' reaction, no epoxide was detected (**Fig. R7**). Thus, H_2O_2 generated by water oxidation can be excluded as the oxidant for the epoxidation of alkene in our system.

In the revised manuscript, the corresponding description (Lines 254-256, Page 6) as “*In addition, the epoxidation in the presence of H_2O_2 shows poor activity on $\alpha\text{-Fe}_2\text{O}_3$ (Fig. S46), excluding the possibility that H_2O_2 generated by water oxidation can act as the oxidant for the epoxidation of alkenes.*” has been added in the revised main text. More detailed description is provided in the revised supplementary information (**Fig. S46**).

Fig. R7 The HPLC spectra obtained after 4h' reaction time of 10 mM **1** with 10 mM H_2O_2 in the dark condition on $\alpha\text{-Fe}_2\text{O}_3$.

4). The authors should add the epoxidation-based incident photon to current efficiencies in the PEC system.

Response: As shown in **Fig. R8**, the monochromatic incident photon-to-electron conversion efficiency (IPCE) experiments were conducted. IPCE measurement showed that the visible-light activity started at ~600 nm and increased with the shortened wavelength. These efficiencies exhibited the maximum values at a wavelength of 400 nm, and increased with increasing of applied biases, which reached to ~21% at 0.55 V vs. Fc/Fc⁺.

To address the reviewer's concern, we have added the above results in the revised manuscript (Lines 116-118, Page 3) as "*The monochromatic incident photon-to-electron conversion efficiency (IPCE) experiments show that the maximum IPCE is ~21% at 0.55 V vs. Fc/Fc⁺ at wavelength of 400 nm (Fig. S6)*". More detailed description is included in the supplementary information (Fig. S6).

Fig. R8 IPCE measures on α -Fe₂O₃ with different applied biases.

5). The TBABr seems act as an pre-mediator for the epoxidation. The possibility of other borides or BrOx-containing compounds to initiate the same reaction should be carefully tested.

Response: According to the reviewer's suggestion, we carried out the PEC

oxidation of alkene in the presence of 3 mM BrO_3^- (approximate saturated solubility). As shown in **Fig. R9a**, the corresponding selectivity and FE of epoxide were 64% and 41%, respectively, indicating that the addition of BrO_3^- species did not obviously influence the epoxidation activity (in TBABF_4 system, selectivity $43\pm 5\%$, FE $41\pm 3\%$). In addition, 20 mM NaBr was used to substitute TBABr to conduct PEC epoxidation reaction (**Fig. R9b**). An excellent selectivity ($> 95\%$) of epoxide was achieved. These results confirmed that Br^- serves as a key mediator to effectively perform alkene epoxidation.

In the revised version, the corresponding description (Lines 127-129, Page 3) of “*When TBABr was replaced by NaBr (Fig. S8), an excellent selectivity ($> 95\%$) of epoxide was also achieved.*” have been added in the main text. More detailed description is provided in the revised supplementary information (under the **Fig. S8**).

Fig. R9 The HPLC spectra obtained at different PEC oxidation time of 10 mM **1** with (a) 100 mM TBABF_4 and 3 mM NaBrO_3 at 0.75 V vs. Fc/Fc^+ and (b) 20 mM NaBr at 0.35 V vs. Fc/Fc^+ on $\alpha\text{-Fe}_2\text{O}_3$.

6). Similarly, the possibility of I^-/IOx^- , or Cl^-/ClO^- is also to be tested to further extend the mediator scope. If NOT, the unique role of Br^-/BrO^- should be well discussed.

Response: According to the suggestion of the reviewer, we performed the PEC oxidation of alkene by using TBACl and TBAI as electrolyte to examine the

possibility of Cl⁻ and I⁻-mediated epoxidation. Contrary to Br⁻-mediator, epoxidation reactions by using both Cl⁻ and I⁻ as the mediator exhibit the poor activity and selectivity (**Fig. R10**). For the TBACl system, the epoxide is detected with poor performance (**Fig. R10a**), while in the TBAI system no oxidation of alkene is observed (**Fig. R10b**). The LSV experiments show that, in the system using TBACl as electrolyte, the negative shift of onset potential and the photocurrent increase relative to the case of TBABF₄ are not very significant (**Fig. R11**), which suggests that the oxidation of Cl⁻ is not very competitive to the water oxidation on α -Fe₂O₃ photoanode under our experimental conditions. Therefore, poor epoxidation performance of the TBACl systems should originate from the unfavorable oxidation of Cl⁻ to ClO⁻. For the TBAI system, the large negative shift of onset potential indicates that I⁻ is easy to be oxidized on the α -Fe₂O₃ photoanode. The low epoxidation activity may be attributed to the low oxidation ability and the poor stability of IO⁻ in the TBAI system. For TBABr system, however, the Br⁻ is facile to be oxygenated to BrO⁻ species on α -Fe₂O₃ under our PEC conditions, and the formed BrO⁻ species is active enough to transfer its oxygen atom to the alkenes. Therefore, the Br/BrO⁻ cycling plays an unique role in mediating the epoxidation.

In the revised version, we have added the above results and discussion in the revised main text (Lines 129-131, Page 4) as “*However, when the Cl⁻ and I⁻ salts of TBA⁺ (TBACl and TBAI) were used, the epoxidation activity had dramatically decreased under the same experimental conditions (Figs. S9-S10), indicative of the unique role of Br⁻ in the epoxidation on α -Fe₂O₃.*” More detailed description and discussion are provided in supplementary information (below **Figs. S9-S10**).

Fig. R10 The HPLC spectra obtained at different PEC oxidation time of 10 mM **1** with 100 mM (a) TBACl or (b) TBAI at 0.15 V vs. Fc/Fc⁺ on α -Fe₂O₃.

Fig. R11 J - V scan of α -Fe₂O₃ with different 100 mM TBAX (X: BF₄⁻, Cl, Br and I) under AM 1.5G illumination measured in CH₃CN solution with 5% H₂O in an Ar atmosphere with 10 mM 4,4'-(CH₃)₂-TSB (**1**). Scan rate 0.05 V s⁻¹.

To Reviewer #2:

In this work, the authors developed an approach to oxidize alkenes to epoxides using Br/BrO⁻ as the mediator for electron transfer. This report is well-written and the authors provided sufficient evidence to show that alkene oxidation is by BrO⁻ rather than Br₂. Isotope experiments further demonstrated that oxygen is from water oxidation.

Response: Thanks for the reviewer's helpful suggestions and comments on our manuscript. We have revised our manuscript according to the reviewer's suggestions.

1. However, a significant drawback of this paper is that this alkene epoxidation only can reach a good faradaic efficiency when alkene contains phenyl functional groups. This severely limits the application of this approach and should be mentioned and explained in the manuscript.

Response: The epoxidation of aromatic alkenes represents an important chemical reaction, and the corresponding aromatic epoxide serves as a versatile mediator for the application of pharmaceuticals. Thus, the synthesis of aromatic epoxide can gain wide application in many fields.

Actually, the selective epoxidation of aliphatic alkenes is quite challenging in most of the developing techniques for epoxidation. **Table R1** lists some recently-reported examples for the epoxidation of aliphatic alkenes. It can be found that, as the commonly employed substrates of aliphatic alkenes, the epoxidation of cyclooctene exhibits the highest FE in these examples. On anode of manganese oxide nanoparticles, the EC epoxidation of cyclooctene exhibits of FE ~ 30% and a selectivity of 72%. The EC epoxidation of cyclooctene on RuO₂/N_{0.12}C anode has a good selectivity but a moderate FE. The selectivity and FE of direct PEC epoxidation of cyclooctene on bare α-Fe₂O₃ are only 9% and 3%, respectively. The selectivity and FE for the epoxidation of other aliphatic alkenes are even poor in these systems. For the Br-mediated EC systems, the FE values for epoxidation of aliphatic alkenes are

in the range of 33-39% (details in SI, entries 6-7, **Table S1**). In our Br-mediated PEC systems, the selectivity and FE for the epoxidation of cyclooctene are 75% and 41%, respectively, which are much higher than those on the bare $\alpha\text{-Fe}_2\text{O}_3$. Moreover, the performance of the Br-mediated epoxidation of aliphatic alkenes is better than or comparable to that on MnO_x and RuO_2 anodes or in the Br-mediated EC and PEC systems.

The relatively mediocre selectivity and FE values for epoxidation of aliphatic alkenes may stem from the relative inertness of C=C bond of aliphatic alkenes toward epoxidation, the reactivity toward other active radical species, and/or the chemical lability of aliphatic epoxide. In our systems, several by-products, including ketones, bromine-substituted products (as shown by GC-MS analysis, **Fig. R12**) are detected.

In the revised manuscript, we have added the corresponding discussion (Lines 201-209, Page 5) as “*It should be noted that, relative to the aromatic alkenes, the selectivity and FE for the epoxidation of aliphatic alkenes is not very efficient, which may stem from the relative inertness of C=C bond of aliphatic alkenes toward epoxidation. Moreover, several by-products, including ketones, bromine-substituted products (as shown by GC-MS analysis, Fig. S43) are detected. Nevertheless, the Br-mediated PEC epoxidation of aliphatic alkenes is still much higher than those on the bare $\alpha\text{-Fe}_2\text{O}_3$. Moreover, the performance of the Br-mediated epoxidation of aliphatic alkenes is better than or comparable to that on MnO_x and RuO_2 anodes or in the Br-mediated EC and PEC systems*” in the revised main text and supplementary information (**Fig. S43**).

Fig. R12 The GC spectra obtained at different PEC oxidation time of 10 mM cyclooctene with 100 mM TBABr at 0.15 V vs. Fc/Fc⁺ on α -Fe₂O₃. The products were analyzed by GC-MS data.

Table R1. Examples for previously reported PEC or EC aliphatic alkene epoxidation.

Entry	Substrate	Reaction conditions	Select. (%)	FE (%)	Ref.
1		PEC on α -Fe ₂ O ₃ TBABr CH ₃ CN/H ₂ O	75±4	41±1	This work
2			87±11	9±2	
3			60±1	6±1	
4			46±3	7±1	
5		PEC on α -Fe ₂ O ₃ TBABF ₄ CH ₃ CN/H ₂ O	9±1	3±1	R1
6		EC on MnO _x TBABF ₄ CH ₃ CN/H ₂ O	72.1	30	R2
7			8.0	2.8	
8			70.1	9.8	
9			54.1	9.4	
10		EC on RuO ₂ /N _{0.12} C	99	40	R3
11			74	-	

12		LiClO ₄	29	-	
13		C ₂ H ₅ OH/H ₂ O	46	-	

R1. Y. Zhao, C. Deng, D. Tang, L. Ding, Y. Zhang, H. Sheng, H. Ji, W. Song, W. Ma, C. Chen, J. Zhao, α -Fe₂O₃ as a versatile and efficient oxygen atom transfer catalyst in combination with H₂O as the oxygen source. *Nat. Catal.* 4, 684-691 (2021).

R2. K. Jin, J. H. Maalouf, N. Lazouski, N. Corbin, D. Yang, K. Manthiram, Epoxidation of Cyclooctene Using Water as the Oxygen Atom Source at Manganese Oxide Electrocatalysts. *J. Am. Chem. Soc.* 141, 6413-6418 (2019).

R3. X. Lin, Z. Zhou, Q. Y. Li, D. Xu, S. Y. Xia, B. L. Leng, G. Y. Zhai, S. N. Zhang, L. H. Sun, G. Zhao, J. S. Chen, X. H. Li, Direct Oxygen Transfer from H₂O to Cyclooctene over Electron-Rich RuO₂ Nanocrystals for Epoxidation and Hydrogen Evolution. *Angew. Chem. Int. Ed.* 61, e202207108 (2022).

2. Another problem that I found is that the authors cited ref 5 as an example of indirect electrochemical epoxidations. However, in the original paper, it was assumed that the reaction was through direct epoxidation. Thus, this should be corrected.

Response: Ref. 5 reports that the presence of Cl⁻ is able to switch off the combustion pathway of ethene to CO₂ on ruthenium-based oxide electrode, and promote the epoxidation reaction channel. In this paper, such a switching effect of Cl⁻ is attributed to the surface reactive sites blocking by adsorbing of Cl⁻ on the surface of the RuO₂ electrode. As pointed out by the reviewer, the epoxidation reaction still occurs through direct epoxidation, but not via a mediated process.

In the revised manuscript, this reference is not cited here.

Overall, this is a good paper, and I support its publication on Nat. Commun.

Response: Thanks for the reviewer's helpful suggestions and comments on our manuscript.

To Reviewer #3:

The authors report a photoelectrochemical route through which epoxidation is mediated by Br, driving ultimate oxygen atom transfer from water to olefins.

Response: We thank the reviewer for reviewing our manuscript, and appreciate your insightful and previous comments and suggestions. We have revised our manuscript according to the suggestions.

1. The authors claim in the introduction that electrochemical routes suffer from poor selectivity (line 38-41). I don't think this is substantiated by the literature. For instance, the following work shows high FE for epoxidation: <https://www.science.org/doi/full/10.1126/science.aaz8459> The work achieves 70% FE for ethylene and propylene. This is one of several examples, others are cited in the paper. The authors should state these results clearly so that the state of the field can be gauged accurately by the reader.

Response: In line 38-41, we tend to describe briefly the general state in the recent reports about indirect electrochemical epoxidation list in Table S1. In the report mentioned by reviewer (*Science*, 2020, **368**, 1228), E. Sargent et al. developed a chloride-mediated electrochemical epoxidation of ethylene and propylene with high performance (with 97% selectivity and 71% FE), by coupling the heterogeneous electrochemical reaction and homogeneous oxidation reaction with an interface, which represents a big progress in the indirect electrochemical epoxidation.

According to the suggestion of the reviewer, we reorganized this part of the introduction (Lines 37-41, Page 1) as "*For example, by coupling the heterogeneous chlorine evolution reaction and homogeneous alkene oxidation reaction with an interface, Leow et al., recently reported the chloride-mediated electrochemical epoxidation of ethylene and propylene can reach 97% selectivity and 71% FE.*" to make the reader easier to understand the relative research field in the revised version.

2. Since these results demonstrate that electrochemically a halogen mediator can be used, it is not clear what problem the introduction of a photon solves. Can the authors articulate specifically why introducing light is important?

Response: In the electrochemical method, the halogen-mediators is oxidized directly by the external bias potential. Accordingly, a high voltage has to be applied for halogen-mediated reaction, which would be energy-extensive consuming. By contrast, in the photoelectrochemical system, the external bias potential is only used to drive the photoinduced conduction band electron to the cathode. The halogen-mediator is oxidized by the photogenerated hole left in the valence band of photoanode ($\text{Fe}^{\text{IV}}=\text{O}$ in the case of $\alpha\text{-Fe}_2\text{O}_3$). Low voltage is enough to achieve the halogen-mediated epoxidation reaction. Therefore, much less electric energy is needed. In other word, in the photoelectrochemical reaction, the oxidation reaction is driven by the light, and the electric bias is only employed to enhance the separation of photoinduced carriers in the photoanode. Thus, the photoelectrochemical reaction provides a promising way to use directly the light energy from solar.

Specifically, on the $\alpha\text{-Fe}_2\text{O}_3$ photoanode, the applied bias in the dark is 1.05 V vs. Fc/Fc^+ to achieve 0.6 mA cm^{-2} , as shown in **Figs. R13** and **S4**. By contrast, only 0.15 V vs. Fc/Fc^+ of bias is needed to obtain the same current under irradiation. In addition, the electrochemical onset potential of bromide oxidation on the Pt electrode (the most common electrode) is $\sim 500 \text{ mV}$ higher than that on $\alpha\text{-Fe}_2\text{O}_3$ with illumination (**Fig. 3**).

According to the reviewer's suggestion, the role of introducing light has been added and reorganized in the introduction part of revised manuscript (Lines 46-48, Page 2) as "Photoelectrochemical (PEC) techniques, *which can utilize photogenerated holes/electrons to achieve chemical conversion and reduce greatly the consumption of electric energy compared with the pure electrochemical methods*".

Fig. R13 J – V scan of α - Fe_2O_3 measured in 0.1 M TBABr solution (CH_3CN with 5% H_2O) under an Ar atmosphere under AM 1.5G illumination (red) and in the dark condition (black).

3. Can the authors conduct an energy balance for pure electrochemical polarization vs the photoelectrochemical route they are conducting? This might highlight potential advantages or disadvantages of their route.

Response: According to the suggestion of the reviewer, we tried to conduct an estimation of the energy balance for the pure electrochemical polarization versus the photoelectrochemical route on α - Fe_2O_3 .

$$\text{Energy balance } (E_b) = \text{Energy Input } (E_i) - \text{Energy Output } (E_o)$$

In our system, energy output refers to the energy consumption from alkene (1) to epoxide (2). Therefore, the values of energy output should be a constant (1 mol of 2), regardless of electrochemical or photoelectrochemical systems. In photoelectrochemical system, the energy input is composed with electric input (E_{ie}) and light input (E_{is}). In these systems, the cost of energy input is determined by the electric consumption, because the cost of light input could be from the solar energy, which is cheap and “green”. The electric consumption (E_{ie}) can be estimated by:

$$E_{ie} = n \times m \times F \times U / \eta_{FE}$$

Where n is the numbers of electrons (2); m is the mole of epoxide (mol); F is faradaic constant (96485 C/mol); U is applied bias (V); η_{FE} is the Faradic efficiency (82%)

The energy difference between the EC and PEC systems is determined by difference in the applied bias (ΔU):

$$\Delta E_{ie} = n \times m \times F \times \Delta U / \eta_{FE} = 211.8 \text{ kJ/mol}$$

At 0.6 mA/cm² of current, the difference in the applied bias between EC and PEC systems is $\Delta U = 0.90$ V. The energy difference is 211.8 kJ/mol, which should be supplied by the light energy.

In the revised version, the role of introducing light has been added and reorganized in the introduction part (Lines 46-48, Page 2) as “Photoelectrochemical (PEC) techniques, *which can utilize photogenerated holes/electrons to achieve chemical conversion and reduce greatly the consumption of electric energy compared with the pure electrochemical methods*”.

That being said, the implementation of a photoelectrochemical scheme is of fundamental interest and will inspire other work in the field.

Response: Thank the reviewer for his/her precious comments and suggestions.

To Reviewer #4:

The authors present the photoelectrochemical (PEC) epoxidation of mostly aryl and diaryl substituted alkenes using α -Fe₂O₃ as photo anode and Pt as cathode and Br⁻ as redox mediator. Overall, the synthetic results for aryl and diaryl substituted alkenes are very good in terms of selectivity and yield with a significant variation in faradaic efficiency, Table 1. The authors claim that hypobromite, BrO⁻, is the oxidising species. I believe this research deserves publication in an organic chemistry oriented journal such as the Journal of Organic Chemistry or Organic Letters but not in Nature Communications

Response: Thank the reviewer for the precious comments and suggestions. The related experiment data and discussions have been added in the revised version. The responses are presented point-by-point as follows.

1. While the synthetic results for aryl and diaryl substituted alkenes are very good, the results for aliphatic cyclic and acyclic alkenes are not with mediocre selectivity (the byproducts were not given) and poor faradaic efficiency.

Response: The epoxidation of aryl and diaryl substituted alkenes represents an important chemical reaction, and the corresponding aromatic epoxide serves as a versatile mediator for the application of pharmaceuticals. Thus, the synthesis of aromatic epoxide can gain a wide application in many fields.

Actually, the selective epoxidation of aliphatic alkenes is quite challenging in most of the developing techniques for epoxidation. **Table R1** shows some recently-reported examples for the epoxidation of aliphatic alkenes. It can be found that, as the commonly employed substrates of aliphatic alkenes, the epoxidation of cyclooctene exhibits the highest FE in these examples. On anode of manganese oxide nanoparticles, the EC epoxidation of cyclooctene exhibits of FE ~ 30% and a selectivity of 72%. The EC epoxidation of cyclooctene on RuO₂/N_{0.12}C anode has a good selectivity but a moderate FE. The selectivity and FE of direct PEC epoxidation of cyclooctene on bare α -Fe₂O₃ are only 9% and 3%, respectively. The selectivity and FE for the

epoxidation of other aliphatic alkenes are even poor in these systems. For the Br-mediated EC and PEC systems, the FE values for epoxidation of aliphatic alkenes are in the range of 33-39% (details in SI, entries 6-7, **Table S1**). In our Br-mediated PEC systems, the selectivity and FE for the epoxidation of cyclooctene are 75% and 41%, respectively, which are much higher than those on the bare α -Fe₂O₃. Moreover, the performance of the Br-mediated epoxidation of aliphatic alkenes is better than or comparable to that on MnO_x and RuO₂ anodes or in the Br-mediated EC and PEC systems.

The relatively mediocre selectivity and FE values for epoxidation of aliphatic alkenes may stem from the relative inertness of C=C bond of aliphatic alkenes toward epoxidation, the reactivity toward other active radical species, and/or the chemical lability of aliphatic epoxide. In our systems, several by-products, including ketones, bromine-substituted products (as shown by GC-MS analysis, **Fig. R12**) are detected.

In the revised version, we have added the corresponding discussion (Lines 201-209, Page 5) as “*It should be noted that, relative to the aromatic alkenes, the selectivity and FE for the epoxidation of aliphatic alkenes is not very efficient, which may stem from the relative inertness of C=C bond of aliphatic alkenes toward epoxidation. Moreover, several by-products, including ketones, bromine-substituted products (as shown by GC-MS analysis, Fig. S43) are detected. Nevertheless, the Br-mediated PEC epoxidation of aliphatic alkenes is still much higher than those on the bare α -Fe₂O₃. Moreover, the performance of the Br-mediated epoxidation of aliphatic alkenes is better than or comparable to that on MnO_x and RuO₂ anodes or in the Br-mediated EC and PEC systems.*” in the revised manuscript and supplementary information (**Fig. S43**).

Fig. R12 The GC spectra obtained at different PEC oxidation time of 10 mM cyclooctene with 100 mM TBABr at 0.15 V vs. Fc/Fc⁺ on α -Fe₂O₃. The products were analyzed by GC-MS data.

Table R1. Examples for previously reported PEC or EC aliphatic alkene epoxidation.

Entry	Substrate	Reaction conditions	Select. (%)	FE (%)	Ref.
1		PEC on α -Fe ₂ O ₃ TBABr CH ₃ CN/H ₂ O	75±4	41±1	This work
2			87±11	9±2	
3			60±1	6±1	
4			46±3	7±1	
5		PEC on α -Fe ₂ O ₃ TBABF ₄ CH ₃ CN/H ₂ O	9±1	3±1	R1
6		EC on MnO _x TBABF ₄ CH ₃ CN/H ₂ O	72.1	30	R2
7			8.0	2.8	
8			70.1	9.8	
9			54.1	9.4	
10		EC on RuO ₂ /N _{0.12} C	99	40	R3
11			74	-	

12		LiClO ₄	29	-	
13		C ₂ H ₅ OH/H ₂ O	46	-	

R1. Y. Zhao, C. Deng, D. Tang, L. Ding, Y. Zhang, H. Sheng, H. Ji, W. Song, W. Ma, C. Chen, J. Zhao, α -Fe₂O₃ as a versatile and efficient oxygen atom transfer catalyst in combination with H₂O as the oxygen source. *Nat. Catal.* 4, 684-691 (2021).

R2. K. Jin, J. H. Maalouf, N. Lazouski, N. Corbin, D. Yang, K. Manthiram, Epoxidation of Cyclooctene Using Water as the Oxygen Atom Source at Manganese Oxide Electrocatalysts. *J. Am. Chem. Soc.* 141, 6413-6418 (2019).

R3. X. Lin, Z. Zhou, Q. Y. Li, D. Xu, S. Y. Xia, B. L. Leng, G. Y. Zhai, S. N. Zhang, L. H. Sun, G. Zhao, J. S. Chen, X. H. Li, Direct Oxygen Transfer from H₂O to Cyclooctene over Electron-Rich RuO₂ Nanocrystals for Epoxidation and Hydrogen Evolution. *Angew. Chem. Int. Ed.* 61, e202207108 (2022).

2. The hypothesis that hypobromite is an interesting one, but it is not proven. It is always somewhat dangerous to carry of mechanistic studies on the most reactive substrate only. Such studies tend to hide other pathways, precisely because of their high reactivity.

Response: As pointed out by the reviewer, the mechanistic studies on the reactive species are challenging, due to the extreme short lifetime and low concentration of the reactive species. In our study, although the direct detection of active species is very difficult, we provide solid experimental evidence that the hypobromite species is the most possible reaction active species. Such a mechanistic assignment is based on the following experimental observations: (1) Much larger onset potential shift (between in TBABr and TBABF₄ systems) on α -Fe₂O₃ than on the radical-involved TiO₂ or Pt electrode was observed (**Fig. 3**), supporting the oxygen transfer pathway but excluding the radical-based mechanism for the Br-mediated epoxidation on α -Fe₂O₃; (2) The participation of BrO⁻ species in the epoxidation was experimentally verified by the detailed comparison in product distributions among different active bromine species (bromine radical, mixture of H₂O and Br₂, NaBrO) under our electrolytic conditions; (3) The difference of product distribution on α -Fe₂O₃ and TiO₂ in Br-mediated epoxidation reactions further confirms the distinguish oxidation pathways.

(a) where does the epoxidation reaction take place? In solution or on the electrode?

Response: The epoxidation reaction should take place in the bulk solution away from the surface of $\alpha\text{-Fe}_2\text{O}_3$. As shown in **Fig. 2a**, the photocurrent in the Br-mediated epoxidation system is nearly unchanged by addition of the alkene substrate. If the mediated reaction takes place on the surface, the regeneration of Br by alkene would increase the local concentration of Br on the surface of the electrode, after adding the alkene substrate. As a result, the photocurrent of Br oxidation should be enhanced. However, no current enhancement by alkene demonstrates that Br-mediated oxidation reaction of alkene takes place in the bulk solution, rather than on the surface of photoanode. By contrast, in the absence of Br, the photocurrent is significantly enhanced by adding alkene, indicating that the direct oxidation of alkene by photoinduced hole occurs on the surface of $\alpha\text{-Fe}_2\text{O}_3$ photoanodes.

In the revised manuscript, we have added the above discussion (Lines 282-288, Page 7) as “*Notably, the photocurrent in the Br-mediated epoxidation system is nearly unchanged by addition of the alkene substrate (Fig. 2a). If the reaction between BrO^- and alkene takes place on the surface, the regeneration of Br by alkene would increase the local concentration of Br on the surface of the electrode, after adding the alkene substrate. As a result, the current of Br oxidation should be enhanced. However, no current enhancement by adding alkene demonstrates that Br-mediated oxidation reaction of alkene takes place in the bulk solution, rather than on the surface of photoanode*”.

Fig. 2a J–V scan of α -Fe₂O₃ under AM 1.5G illumination measured in 100 mM TBABF₄ (black lines) and TBABr (red lines) solution (CH₃CN with 5% H₂O) under an Ar atmosphere without (dash) and with (solid) 10 mM 4,4'-(CH₃)₂-TSB (1). Scan rate 0.05 V s⁻¹. The dotted vertical lines indicate the potentials (0.15 and 0.75 V vs. Fc/Fc⁺) of photoelectrolysis experiments.

(b) How is hypobromite stable at pH -3, as the results in Table S3 suggests when the pKa of HOBr is 7.5.

Response: In our system, the PEC reaction was carried out in CH₃CN solution with typically only 5 vol% H₂O as oxygen source. Considering that the term “pH” is not applicable in the non-aqueous solution, the pH value listed in **Table S4** refers to the pH of the added water to reflect relative acidity/basicity of solution, but not the pH of the whole CH₃CN solution. Under our experimental conditions, the stability of hypobromite and its pH-dependency, which may be largely different from those in the aqueous solution, is unclear. Nevertheless, the systematic comparisons on the product distribution of alkene oxidation by different Br active species (Br₂, Br • and BrO⁻) under our solution conditions (**Figs. 3d-3e** and **S45**), suggest that BrO⁻ is the most possible active species for the epoxidation reaction.

In order to avoid misleading, we have labelled the pH values referred to

the 5 vol% content of water in the revised manuscript and added the detailed discussion (under the Table S4) in the revised supplementary information.

3. The authors use "solar light" for the experiments. Which wavelengths are relevant? What is the quantum yield?

Response: "Solar light" for our experiments is simulated by a 500 W Xe lamp with an AM 1.5G filter and the light intensity of 100 mW cm^{-2} . For the $\alpha\text{-Fe}_2\text{O}_3$ photoanode, due to its bandgap is about 2.1 eV (Fig. R1b, UV-vis diffuse spectra), the wavelengths of $\sim 600 \text{ nm}$ are relevant which can be used to excite the $\alpha\text{-Fe}_2\text{O}_3$ photoanode to obtain the photo-generated holes/electrons.

The IPCE measurement (Fig. R8) showed that the visible-light activity started at $\sim 600 \text{ nm}$ and increased with the shortened wavelength. The efficiency exhibits a maximum value at a wavelength of 400 nm , and increased with increasing of applied biases, which reached $\sim 21\%$ at 0.55 V vs. Fc/Fc^+ .

To address the reviewer's concern, we have added the above result (Lines 116-118, Page 3) as "The monochromatic incident photon-to-electron conversion efficiency (IPCE) experiments show that the maximum IPCE is $\sim 21\%$ at 0.55 V vs. Fc/Fc^+ at wavelength of 400 nm (Fig. S6)." in the revised manuscript and supplementary information (Figs. S2b and S6).

Fig. R1b UV-vis diffuse spectrum of $\alpha\text{-Fe}_2\text{O}_3$ photoanodes.

Fig. R8 IPCE measures on $\alpha\text{-Fe}_2\text{O}_3$ with different applied biases.

4. The epoxidation appears to be the result of an electrophilic attack of BrO^- on the double bond but this should be demonstrated through use of kinetics and Hammett plots. See Chemical Science, 2020, 11, 11584 – 11591 as a example.

Response: According to the reviewer's suggestions, a corresponding Hammett plots was obtained by the reaction kinetics of epoxidation of different alkenes. As shown in **Fig. R14**, the Hammett linear free-energy relationship shows a negative slope of -0.20 , which indicates an electrophilic attack of BrO^- on the double bond.

In the revised version, the results and discussions of Hammett plots (Lines 277-280, Page 7) as *“Based on the epoxidation reaction kinetics of various para-substituent Trans-stilbene substrates, the Hammett linear free-energy relationship was calculated and showed a negative slope of -0.20 (Fig. S48), which confirms an electrophilic attack of BrO^- on the double bond of alkenes.”* have been added to support our assumption in the revised manuscript and supplementary information (**Fig. S48**).

Fig. R14 The Hammett plot with respect to σ_p values for the PEC epoxidation with various para-substituent *Trans*-stilbene substrates.

5. The reaction also occurs in the absence of Br⁻, Figure 2c albeit at somewhat higher potentials. How do these reactions proceed, how? at what selectivity? Has

Response: In the **Fig. 2c**, the selectivity and FE for the epoxide product without Br⁻ were only 43±5% and 41±3%, which are much lower than that in Br⁻-mediated system. This epoxidation of alkenes in the absence of Br⁻ proceeds by the direct photoelectrochemical oxidation. As shown in our previous study^{R1} (**Fig. R15**), the epoxidation can be achieved through the oxygen transfer from the high-valence iron-oxo (Fe^{IV}=O) species to the alkenes. However, the poor compatibility between the hydrophilicity of hematite surface and the low polarity of C=C bond disfavors their interaction, and results in poor selectivity and FE.

Fig. R15 The proposed direct PEC epoxidation mechanism in TBABF₄ system.

6. In some of the GC plots it looks like there are two products that are poorly separated. Why is that?

Response: It is clear that the GC plots with asymmetric peak come from the product oxide, and the asymmetry should result from the high polarity of these products. To confirm that the GC peak with broad width represents only one product, the corresponding GC plots are obtained under the identical GC conditions by using the standard sample at different concentrations. As depicted in **Fig. R16**, the widening of GC peaks is also observed at all sample concentrations, and good linearity between the peak area and sample concentration is obtained, confirming that the broad peak only represents one product rather than two poorly separated products.

To address the reviewer's concern, we have added the corresponding description as "*The widening of GC peaks is also observed at all sample concentrations, and good linearity between the peak area and sample concentration is obtained, confirming that the broad peak only represents one product rather than two poorly separated products. Such a widening of GC peak may stem from the high polarity of product oxide.*" in the revised supplementary information (under the **Fig. S37b**).

Fig. R16 The GC spectra of cyclooctene oxide with different concentrations in 0.1 M TBABr acetonitrile and water (5 vol%) mixture solution.

7. The authors should also disclose the possibility of formation of benzylic cations or radical cations, followed by nucleophilic attack. CV's of some substrates would be a good start.

Response: According to the suggestion of the reviewer, we examined the CV curves for the PEC epoxidation of 10 mM substrate **1** on α -Fe₂O₃ measured in the presence of 0.1 M TBABr solution. As shown in **Fig. R17**, no obvious reversible redox peak was observed, which demonstrates that benzylic cations or radical cations are not involved in the epoxidation process.

According to the reviewer's suggestion, we have added the above result and discussion (Lines 280-282, Page 7) as "**Moreover, no obvious reversible redox peak was detected (Fig. S49), suggesting that benzylic cations or radical cations are not involved in the epoxidation process.**" in the revised main text and supplementary information (**Fig. S49**).

Fig. R17 The CV curve of $\alpha\text{-Fe}_2\text{O}_3$ measured in 0.1 M TBABr solution (CH_3CN with 5% H_2O) with 10 mM substrate **1** in an Ar atmosphere under AM 1.5G illumination.

Summary - An organic chemistry oriented journal such as the Journal of Organic Chemistry or Organic Letters is more reasonable for this research as it stands. Nature Communications would require a significant amount of additional research.

Response: Thank the reviewer for his/her precious comments and suggestions. After modification, the manuscript has been essentially improved under the suggestions of the reviewers. We believe that it should reach the requirement of Nature Communications now.

REVIEWERS' COMMENTS

Reviewer #1 (Remarks to the Author):

The authors have provided detailed responses to the reviewers' comments. Unfortunately, an obvious technical defect of this paper has been presented in the Figure S2e and Figure S2f. It is NOT acceptable to provide the same TEM image of a particle to confirm the stability of the spent catalyst. SO, I could not suggest the publication of the revised version of the manuscript on Nature Communication.

Reviewer #2 (Remarks to the Author):

The added literature context, energy balance, and discussion of the photon's effect make this manuscript suitable for publication.

Response to Reviewer's Comments

(Reviewer's comments in black font. Our response in blue font. *Revision in manuscript in yellow background.*)

Thanks to all reviewers' valuable comments and suggestions on the manuscript of *NCOMMS-22-38942A* submitted to the journal of ***Nature Communications***.

To Reviewer #1:

The authors have provided detailed responses to the reviewers' comments. Unfortunately, an obvious technical defect of this paper has been presented in the Figure S2e and Figure S2f. It is NOT acceptable to provide the same TEM image of a particle to confirm the stability of the spent catalyst. SO, I could not suggest the publication of the revised version of the manuscript on Nature Communication.

Response: Thanks for the reviewer's kind reminder and careful check on our revised manuscript. We have made a mistake to use the same TEM picture in Figures S2e and S2f. We are so sorry for such carelessness. The wrong TEM image in Figure S2f has been replaced by a correct one (Figure R1f) in the new revised version.

More detailed descriptions are provided in the revised supplementary information (*Figs. S2*).

Fig. R1 The characterizations of fresh and used α - Fe_2O_3 photoanodes. (a) XRD spectra; (b) UV-vis diffuse spectra; SEM images of fresh (c) and used (d) α - Fe_2O_3 ; TEM images of fresh (e) and used (f) α - Fe_2O_3 .

To Reviewer #2:

The added literature context, energy balance, and discussion of the photon's effect make this manuscript suitable for publication.

Response: Thanks for the reviewer's precious suggestions. We have added the corresponding literatures of energy balance and photon's effect in the revised manuscript. The discussion of the photon's effect has also been added in the revision. More details please see the references 27-28, main text (Page 3, Line 25-28) and Supplementary information.